# Crystal structures of U6 snRNA-specific terminal uridylyltransferase

Seisuke Yamashita[1], Yuko Takagi[2], Takashi Nagaike[1] & Kozo Tomita[1]

The terminal uridylyltransferase, TUT1, builds or repairs the 3′-oligo-uridylylated tail of U6 snRNA. The 3′-oligo-uridylylated tail is the Lsm-binding site for U4/U6 di-snRNP formation and U6 snRNA recycling for pre-mRNA splicing. Here, we report crystallographic and biochemical analyses of human TUT1, which revealed the mechanisms for the specific uridylylation of the 3′-end of U6 snRNA by TUT1. The $O_2$ and $O_4$ atoms of the UTP base form hydrogen bonds with the conserved His and Asn in the catalytic pocket, respectively, and TUT1 preferentially incorporates UMP onto the 3′-end of RNAs. TUT1 recognizes the entire U6 snRNA molecule by its catalytic domains, N-terminal RNA-recognition motifs and a previously unidentified C-terminal RNA-binding domain. Each domain recognizes specific regions within U6 snRNA, and the recognition is coupled with the domain movements and U6 snRNA structural changes. Hence, TUT1 functions as the U6 snRNA-specific terminal uridylyltransferase required for pre-mRNA splicing.

[1] Department of Computational Biology and Medical Sciences, Graduate School of Frontier Sciences, the University of Tokyo, Kashiwa, Chiba 277-8562, Japan. [2] National Institute of Advanced Industrial Science and Technology, Biomedical Research Institute, 1-1-1 Higashi, Tsukuba, Ibaraki 305-8566, Japan. Correspondence and requests for materials should be addressed to K.T. (email: kozo_tomita@cbms.k.u-tokyo.ac.jp).

Pre-mRNA splicing in eukaryotic cells is a series of reactions catalysed by a large ribonucleoprotein (RNP) complex, the spliceosome. The major spliceosome comprises five small ribonucleoprotein complexes (U1, U2, U4, U5 and U6 snRNPs) and a large number of proteins[1]. The U6 snRNP is composed of the U6 snRNA, and the p110 (hPrp24) and heteroheptameric Lsm2-8 proteins. The U6 snRNP enters the splicing cycle through the formation of the U4/U6 di-snRNP. p110 and Lsm2-8 catalyse the annealing of the U6 and U4 snRNAs[2–4], leading to the formation of the U4/U6 di-snRNP, which is followed by the U4/U6 · U5 tri-snRNP formation. The U4/U6 · U5 tri-snRNP is recruited into the pre-spliceosome, composed of the pre-mRNA and the U1 and U2 snRNPs. The U6 snRNA forms an alternative helix with the U2 snRNA, and the two splicing reaction steps proceed with the structural rearrangements of the U6 snRNA in the spliceosome. In the base-paired U6-U2 snRNAs, the U6 snRNA participates in the active site formation and the divalent cation coordination for the catalysis of the trans-esterification reactions[5].

The U6 snRNA is transcribed by RNA polymerase III, and undergoes multiple processing steps before entering the splicing cycles (Supplementary Fig. 1)[6]. The U6 snRNA transcript has 5′-stem, internal stem-loop (ISL) and telestem secondary structures[7,8]. The primary transcript of the U6 snRNA has four genome-encoded uridines at its 3′-end. After transcription, the 3′-end is oligo-uridylylated by a terminal uridylyltransferase, TUT1 (TUTase6)[9,10], one of the members of the non-canonical nucleotidyltransferases[11–13]. Subsequently, the oligo-uridylylated tail is trimmed by a 3′-5′ exonuclease, Mpn1 (Usb1)[14–16], and the 3′ end of the resultant mature U6 snRNA has five uridines with a 2′,3′-cyclic phosphate ($>$p). These 3′-maturation processes of the U6 snRNA protect it from degradation. The oligo-uridylylated tail of the U6 snRNA is the binding site for the Lsm2-8 ring protein complexes[4,17], for the annealing of the U6 and U4 snRNAs to form the di-U4/U6 snRNP, and for the recycling of the U6 snRNA after the splicing reaction, together with p110 (ref. 18). Thus, the 3′-oligo-uridylylated tail of U6 snRNA contributes to efficient pre-mRNA splicing in cells.

Here, we present the crystal structures of human TUT1, and its complexes with UTP or ATP. Crystallographic and biochemical studies of TUT1 revealed the molecular mechanism underlying the specific oligo-uridylylation of the 3′-end of U6 snRNA by TUT1.

## Results

**Structure determination of human TUT1.** Our extensive trials to obtain crystals of full-length human TUT1 were unsuccessful. Therefore, the amino-terminal putative zinc finger (ZF) and RNA recognition motif (RRM) (amino-acid residues 1–140), the predicted disordered proline-rich region (PRR: amino-acid residues 235–304) and the carboxy-terminal region (amino-acid residues 651–750) neighbouring the nuclear localization signal (NLS) were removed[9] (Fig. 1a, Supplementary Fig. 2). The truncated TUT1 (TUT1_ΔN, Fig. 1a) was overexpressed in *Escherichia coli*, purified and crystallized. Finally, X-ray-diffracting crystals of TUT1_ΔN were obtained. To improve the quality of the crystals, several cysteine residues in TUT1_ΔN were further replaced with alanines or serines. Three crystal forms (form-I, -II and -III) were obtained, and the structures were analysed (Table 1).

The form-I and -II crystals belong to the space groups $P2_12_12$, and $P2_1$, respectively. The form-I TUT1_ΔN structure complexed with UTP (soaked in) was initially solved by the single isomorphous replacement with anomalous scattering (SIRAS) method, using the protein complexed with UTP in the presence

of $BaCl_2$ (Supplementary Fig. 3). Subsequently, the structure was modelled and refined to an $R$ factor of 24.6% ($R_{free}$ of 29.4%), using reflections up to 2.95 Å resolution. The data sets were also collected from the *apo* form-I crystal, and the structure was refined to an $R$ factor of 25.3% ($R_{free}$ of 29.7%), using reflections up to 2.8 Å resolution. The structures of form-II TUT1_ΔN with UTP and ATP soaked in were also analysed, and the structures were modelled and refined to $R$ factors of 22.9% ($R_{free}$ of 25.7%) and 24.1% ($R_{free}$ of 28.2%) for the UTP complex and the ATP complex, using reflections up to 2.7 and 2.8 Å resolutions, respectively. The form-III crystal belongs to the space group $P4_32_12$, and its structure was also determined and refined to an $R$ factor of 21.4% ($R_{free}$ of 25.8%), using reflections up to 3.4 Å resolution.

The crystal of TUT1 lacking the N-terminal Zn-finger, PRR and C-terminal region (TUT1_ΔC, Fig. 1a) was also obtained (form-IV). The crystal belongs to the space group $P4_32_12$. The structure was modelled and refined to an $R$ factor of 20.3% ($R_{free}$ of 24.2%) up to 3.2 Å resolution. The catalytic activities of the TUT1 variants used for crystallization were examined. TUT1_ΔN and TUT1_ΔC (Fig. 1a) have lower uridylylation activities than wild-type TUT1 (Supplementary Fig. 4), suggesting the importance of the N-terminal ZF and RRM and the C-terminal KA-1 (kinase associated-1) domain, as described below.

**Overall structure of human TUT1.** The TUT1_ΔN structure consists of three domains—catalytic palm (residues 172–403) and fingers (residues 141–171 and 404–598) domains, and an additional distinct domain linked to the C-terminus of the protein (residues 599–874) (Fig. 1a, Supplementary Fig. 2). The C-terminal region of TUT1 is a previously unidentified RNA-binding domain, and it was named the KA-1 domain, as described below.

The palm domain consists of five-stranded β-sheets (β1–β5) and two α-helices (α3 and α4), and a catalytic triad (Asp216, Asp218 and Asp381) resides in the domain. The PRR, which was removed from the TUT1 protein for crystallization, resides between β6 and α6 in the palm domain. The palm domain structure of TUT1 is homologous to those of the DNA polymerase β family of proteins. The fingers domain has a helical structure with ten α-helices (α1–α2, α5–α12) and three β-sheets (β6–β8), and is homologous to the central domain of PAPα (refs 19,20). The nucleotide is located in the cleft between the palm and fingers domains (Fig. 1a). The overall structure of the catalytic core palm and fingers domains of TUT1 is topologically homologous to those of the yeast Cid1 and vertebrate mitochondrial PAP (PAPD1) proteins[21–24] (Fig. 2a).

The C-terminal domain of TUT1 consists of four anti-parallel β-sheets (β8–β11) and five α-helices (α14–α18) (Fig. 1a,b). The four-stranded β-sheets (β8–β11) are preceded by α15, and are followed by α18. The region encompassing the NLS, which was removed for crystallization, resides between β8 and β9. A BLAST homology search of the C-terminal primary amino-acid sequence of TUT1 did not generate any relevant homologous proteins or domains. However, a structural homology search using the Dali server[25] revealed that a portion of this domain (α15, β8–β11 and α18) is topologically homologous to the KA-1 domain from various proteins [$Z$-score of 8.3 for Map/Microtubule Affinity-Regulating Kinase 3 (MARK-3)] (Fig. 1b)[26]. The possible function of the KA-1 domain of TUT1 is described below.

The form-III TUT1_ΔN adopts a closed conformation, as compared with the form-I TUT1_ΔN. A comparison between the form-I and -III TUT1_ΔN structures highlights the mobility of

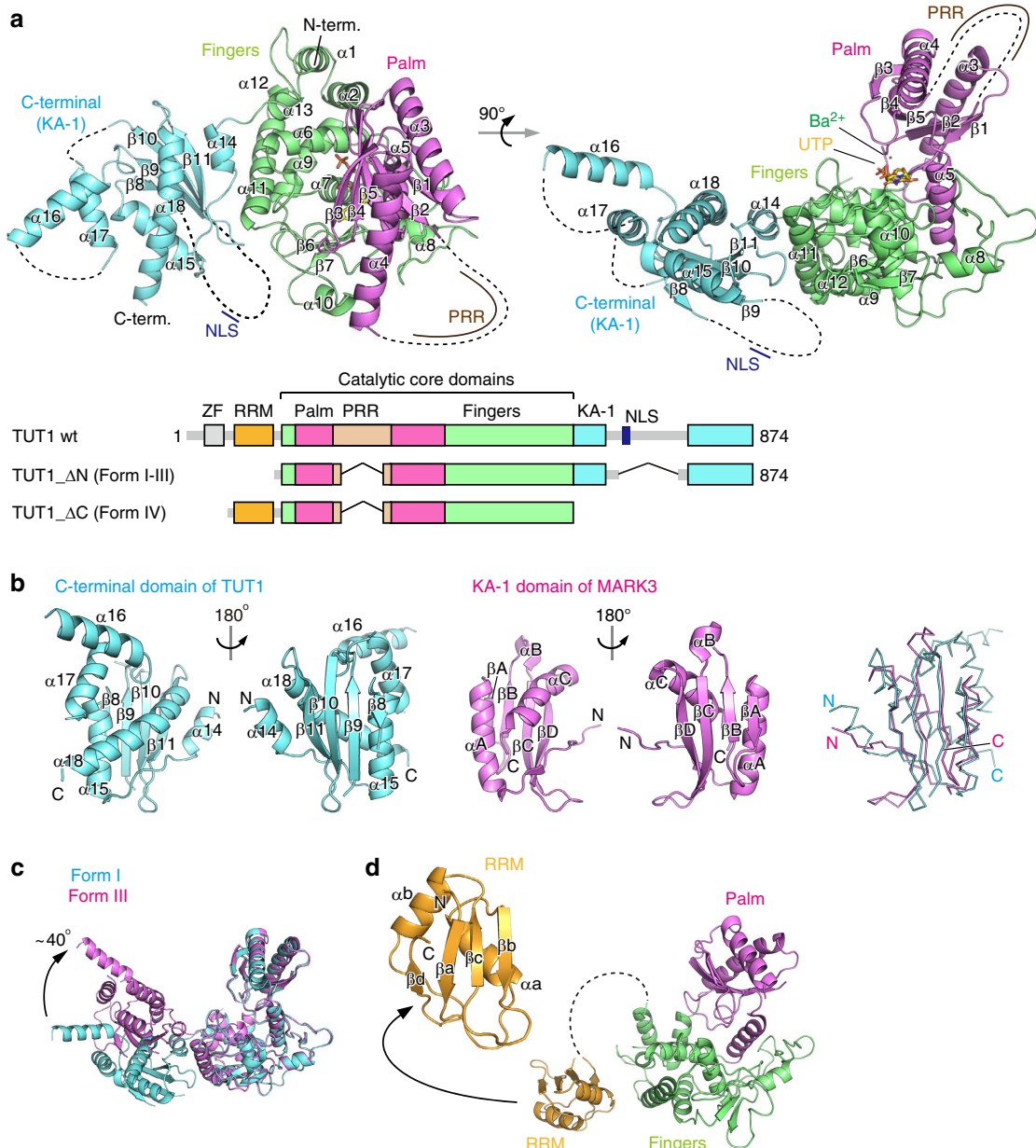

**Figure 1 | Overall structure of human TUT1.** (**a**) Overall structure of form-I TUT1_ΔN. The palm, fingers and C-terminal (KA-1) domains are coloured magenta, green and cyan, respectively. Dashed lines are the proline-rich and nuclear localization signal regions removed for crystallization, and the disordered regions. UTP is depicted by a yellow stick model. The barium ion (Ba²⁺) is depicted by a dark green sphere. Schematic diagrams of human full-length TUT1, TUT1_ΔN and TUT1_ΔC, which were crystallized. ZF, zinc finger; RRM, RNA recognition motif; PRR, proline-rich region; NLS, nuclear localization signal; KA-1, kinase associated-1. (**b**) Comparison of the structures of the C-terminal domain of TUT1 (left, cyan) and the KA-1 domain of MARK-3 (ref. 26) (middle, magenta), and superimposition of the two structures in ribbon models. (**c**) Superimposition of the form-I (cyan) and -III (magenta) structures of TUT1_ΔN. The form-III structure adopts a closed form relative to the form-I structure, with respect to the catalytic core domains. (**d**) Overall structure of TUT1_ΔC (form-IV). The RRM is coloured orange.

the KA-1 domain. The KA-1 domain rotates by ∼40° with respect to the catalytic core domains, using α14 as the axis of rotation (Fig. 1c).

In the form-IV TUT1_ΔC structure, the N-terminal domain adopts a typical RRM fold[27], with four anti-parallel β-sheets stacked onto two α-helices (Fig. 1d). A comparison of the structures of the N-terminal RRM of TUT1 and the U2AF65 RRM1 suggested that the conserved aromatic or hydrophobic resides (Phe59, Phe94 and Ile96) on the β-strands would be involved in single-stranded RNA binding[28] (Supplementary

Fig. 5). The linker region (residues 127–144), bridging the RRM and the fingers, was disordered and the electron density was not observed. Since the RRM does not interact with other domains in TUT1_ΔC in the crystal structure, the N-terminal RRM would be mobile in the substrate RNA-free form of TUT1.

**Nucleotide specificity of human TUT1.** Human TUT1 was originally identified as a U6 snRNA-specific terminal uridylyltransferase[9]. Subsequently, it was also reported that TUT1

**Table 1 | Data collection and refinement statistics.**

| | Form-I *apo* | Form-I BaUTP | Form-II MgUTP | Form-II MgATP | Form-III *apo* | Form-IV *apo* |
|---|---|---|---|---|---|---|
| *Data collection* | | | | | | |
| Space group | $P2_12_12_1$ | $P2_12_12_1$ | $P2_1$ | $P2_1$ | $P4_32_12$ | $P4_32_12$ |
| Cell dimensions | | | | | | |
| *a, b, c* (Å) | 77.63, 87.78, 185.07 | 77.33, 88.40, 184.19 | 79.79, 85.11, 93.76 | 77.66, 88.56, 93.82 | 173.14, 173.14, 208.58 | 142.53, 142.53, 282.69 |
| α, β, γ (°) | 90, 90, 90 | 90, 90, 90 | 90, 99.57, 90 | 90, 98.66, 90 | 90, 90, 90 | 90, 90, 90 |
| Wavelength (Å) | 0.98000 | 1.5000 | 0.98000 | 0.98000 | 0.98000 | 0.98000 |
| Resolution (Å)*,† | 50-2.8 (2.90-2.80) | 50-2.95 (3.05-2.95) | 50-2.7 (2.80-2.70) | 50-2.8 (2.90-2.80) | 50-3.4 (3.53-3.40) | 50-3.2 (3.32-3.21) |
| $R_{sym}$*,† | 0.188 (1.866) | 0.258 (2.012) | 0.084 (0.700) | 0.134 (1.448) | 0.334 (1.342) | 0.375 (2.225) |
| $I/\sigma I$*,† | 10.2 (1.3) | 15.4 (2.3) | 12.3 (1.9) | 13.1 (1.4) | 8.7 (2.3) | 14.3 (2.0) |
| $CC_{1/2}$*,† | 0.998 (0.635) | 0.998 (0.839) | 0.997 (0.635) | 0.998 (0.530) | 0.993 (0.751) | 0.995 (0.694) |
| Completeness (%)*,† | 99.8 (98.6) | 99.9 (99.5) | 99.2 (97.7) | 99.8 (98.9) | 89.8 (37.0) | 99.9 (99.2) |
| Redundancy*,† | 9.5 (9.1) | 28.8 (28.7) | 3.4 (3.4) | 6.8 (6.6) | 13.0 (12.3) | 26.4 (26.4) |
| | | | | | | |
| *Phasing* | | | | | | |
| Ba sites | | 2 | | | | |
| Figure of merit | | 0.280 | | | | |
| | | | | | | |
| *Refinement* | | | | | | |
| Resolution (Å) | 20-2.8 | 20-2.95 | 20-2.7 | 20-2.8 | 20-3.4 | 20-3.2 |
| No. of reflections | 31,837 | 27,435 | 33,814 | 31,113 | 39,562 | 48,397 |
| $R_{work}/R_{free}$ | 25.29/29.69 | 24.60/29.38 | 22.88/25.69 | 24.10/28.24 | 21.40/25.82 | 20.28/24.18 |
| No. of atoms | | | | | | |
| Protein | 7,567 | 7,522 | 7,527 | 7,527 | 15,026 | 12,187 |
| Ligand/ion | 2 | 62 | 70 | 66 | – | 7 |
| Water | – | – | 2 | 2 | – | – |
| *B*-factors (Å²) | | | | | | |
| Protein | 76.8 | 68.9 | 68.1 | 82.0 | 58.90 | 73.70 |
| Ligand/ion | 73.1 | 40.1 | 72.0 | 162.0 | – | 54.30 |
| Water | – | – | 38.2 | 92.8 | – | – |
| R.m.s.d's | | | | | | |
| Bond lengths (Å) | 0.003 | 0.006 | 0.007 | 0.003 | 0.003 | 0.003 |
| Bond angles (°) | 0.79 | 0.89 | 1.02 | 0.89 | 0.84 | 0.77 |

*Values in parentheses are for highest-resolution shell.
†Data set of form-III *apo* crystal was anisotropically scaled and truncated to 3.4, 3.4 and 3.8 Å resolutions along the *a*\*, *b*\* and *c*\* axes, respectively.

can function as a poly(A) polymerase under specific conditions[29]. Therefore, the complex structures of TUT1_ΔN with either UTP or ATP soaked in were analysed (Fig. 2b,c).

Both UTP and ATP reside in the cleft between the palm and fingers domains (Figs 1a and 2b,c). In the structure of UTP-bound TUT1_ΔN, one magnesium ion is coordinated by the β-γ phosphates of UTP and the catalytic Asp216 and Asp218 residues. The phosphate group of UTP forms hydrogen bonds with the Nη atom of Arg414, the Nδ atom of Asn432 and the main-chain amino-group of Asp216. The uracil base is sandwiched between Tyr432 and the side chain of Arg366. The $O_2$ and $O_4$ atoms of UTP form hydrogen bonds with the Nδ atom of Asn392 and the Nε atom of His549, respectively. The $N_3$ atom of UTP forms a hydrogen bond with a water molecule that also hydrogen bonds with Asp543 (Supplementary Fig. 6). The ribose 2′-OH of UTP forms hydrogen bonds with the Oδ atom of Asn392, thus discriminating the ribose from the deoxyribose (Fig. 2b). In the ATP-bound structure, the electron density of ATP is weaker than that of UTP, and only the $N_1$ atom of the adenine base of ATP hydrogen bonds with the Nε atom of His549 (Fig. 2c), and the ribose 2′-OH group of ATP hydrogen bonds with the Oδ atom of Asn392, as also observed in the complex structure of yeast Cid 1 with dATP[21] (Supplementary Fig. 6). Superimpositions of the structures of TUT1 complexed with UTP (Fig. 2b,c) and yeast Cid1 complexed with ApU[30], and the positions of UTP (or ATP) and $Mg^{2+}$ relative to the catalytic carboxylates (Asp216 and Asp218 and Asp381) in TUT1, suggest that UTP (or ATP) binds within the incoming nucleotide site in the catalytic pocket (Supplementary Fig. 6e).

The nucleotide specificities of TUT1 were analysed, using a U6 snRNA transcript with four 3′-Us (U6 snRNA-u4) (Fig. 2d). TUT1 incorporated UMP more efficiently than AMP into U6 snRNA-u4. The steady-state kinetics of nucleotide incorporations into U6 snRNA-u4 showed estimated $K_m$ values for ATP and UTP of 1,380 and 59 μM, respectively, and estimated $k_{cat}$ values for ATP and UTP of 0.002 and 0.059 s$^{-1}$, respectively (Fig. 2e). Thus, UTP is a much better substrate than ATP (around 700-fold) for TUT1 *in vitro*. The mechanism of nucleotide recognition by TUT1 and the specificity of TUT1 are essentially the same as those of yeast Cid1 (refs 21–23).

Together with the present complex structures of TUT1 with nucleotides, TUT1 preferentially incorporates UMP onto the 3′-end of the U6 snRNA. Although TUT1 can also function as a polyA polymerase under specific conditions and in certain biological processes[29,31–33], the present structural and biochemical studies indicate that TUT1 is an intrinsic terminal uridylyltransferase.

**Domain requirement for uridylylation of U6 snRNA.** The *in vitro* biochemical results described above and previous reports showed that TUT1 specifically uridylylates the 3′-end of the U6 snRNA[9,10,34]. To explore the mechanisms by which TUT1 uridylylates the U6 snRNA specifically, truncated TUT1 variants were tested for their uridylylation activity on the 3′-end of the U6 snRNA *in vitro*. As compared with the structure of yeast Cid1, human TUT1 has additional domains. TUT1 is composed of N-terminal Zn-finger, RRM, palm, fingers and KA-1 domains (Figs 1a and 3a and Supplementary Fig. 2). PRR is inserted in the middle of the palm domain. The domain organization of TUT1 is also different from those of other human terminal uridylyltransferase (TUTase) families[11–13].

Analyses of *in vitro* UMP incorporation into U6 snRNA-u4 revealed that TUT1 variants lacking the N-terminal Zn-finger (ΔZ) and lacking both the Zn-finger and RRM (ΔZR) domains have lower uridylylation activity, while TUT1 variants lacking the

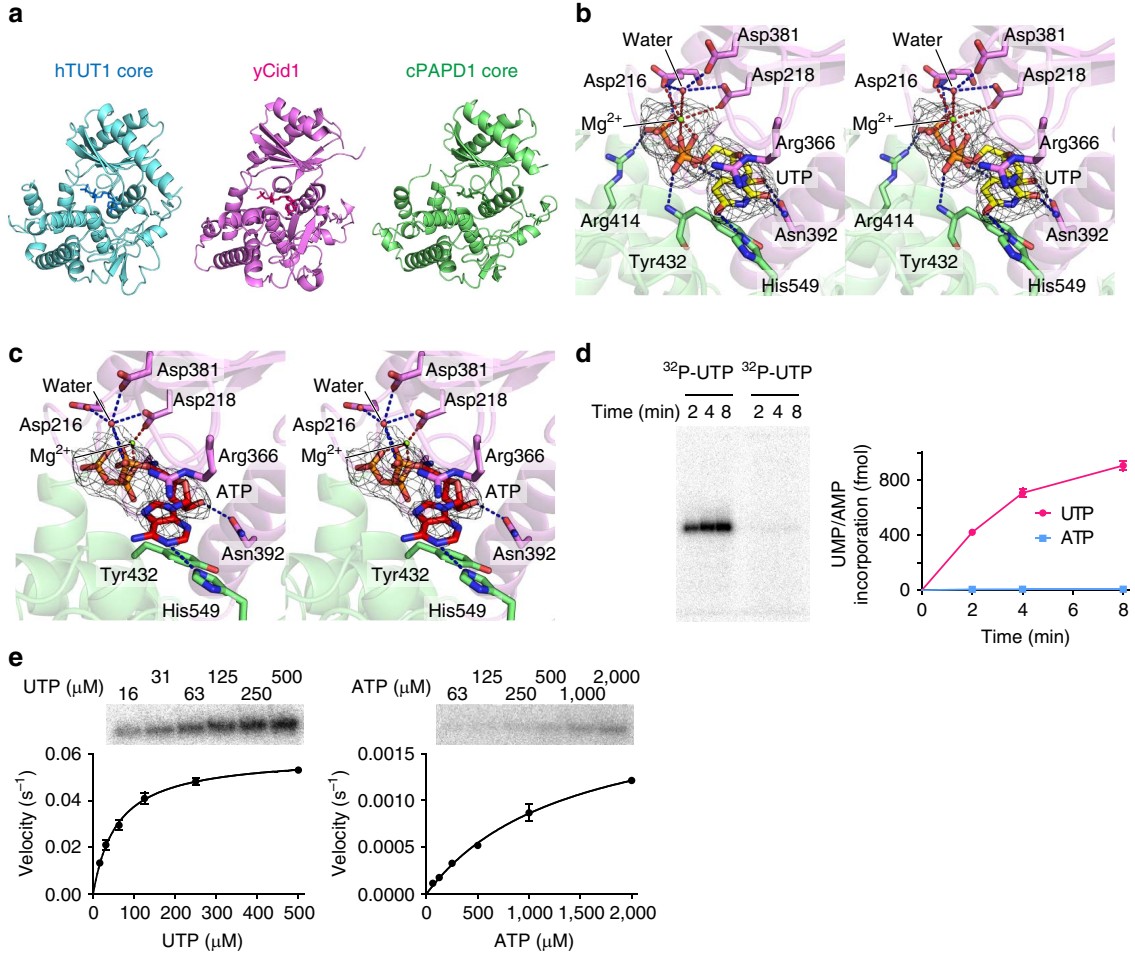

**Figure 2 | Nucleotide recognition by human TUT1.** (**a**) Structure of the catalytic domain of human TUT1. Comparison of the catalytic core domains of human TUT1 (amino-acid residues 145–598; left, cyan) with those of yeast Cid1 (amino-acid residues 42–381; middle, magenta)[21–23] and vertebrate mitochondrial PAP (PAPD1) (amino-acid residues 163–245 and 270–527; right, green)[24,37]. (**b**) UTP recognition by human TUT1. $F_o$–$F_c$ omit map of UTP (yellow stick model) contoured at $3.0\sigma$ (coloured grey). (**c**) ATP recognition by human TUT1. $F_o$–$F_c$ omit map of ATP (red stick model) contoured at $3.0\sigma$ (coloured grey). (**d**) Time-courses of UMP or AMP incorporation onto the 3′-end of the U6 snRNA-u4 transcript. (**e**) Steady-state kinetics of UMP (left) and AMP (right) incorporation into the U6 snRNA-u4 transcript, with various UTP (0–500 μM) and ATP (0–2,000 μM) nucleotide concentrations. Bars in the graphs indicate s.d. of two independent experiments.

PRR (ΔPRR) and lacking the KA-1 domain (ΔKA-1) have almost the same activity as wild-type TUT1, under standard conditions (1.0 μM of U6 snRNA-u4, Fig. 3b).

Analyses of the steady-state kinetics of nucleotide incorporation into U6 snRNA-u4 by wild-type TUT1 and its variants generated estimated $K_m$ values of U6 snRNA-u4 for wild-type TUT1, ΔPRR and ΔKA-1 of 55, 55 and 558 nM, respectively (Fig. 3c). The $k_{cat}$ values for the uridylylation of U6 snRNA for wild-type TUT1, ΔPRR and ΔKA-1 are estimated to be 0.061, 0.058 and 0.13 s$^{-1}$, respectively. The overall uridylylation efficiencies ($k_{cat}/K_m$) of ΔPRR and ΔKA-1 are 96 and 20% of that of wild-type TUT1. Thus, the C-terminal KA-1 domain increases the affinity towards the U6 snRNA at the UMP incorporation stage. While wild-type TUT1 and ΔPRR rapidly add three to four UMPs onto U6 snRNA-u4, ΔKA-1 adds two to three UMPs under the same conditions (Supplementary Fig. 7a).

The $K_m$ values of U6 snRNA for ΔZ and ΔZR are estimated to be 341 and 377 nM, respectively, and the $k_{cat}$ values for the uridylylation of U6 snRNA by ΔZ and ΔZR are estimated to be 0.0006 and 0.0003 s$^{-1}$, respectively (Fig. 3d). The overall uridylylation efficiencies of ΔZ and ΔZR are less than 0.2% of that of wild-type TUT1, and their lower activities arose from the reduced catalytic efficiencies. Thus, the N-terminal Zn-finger and RRM domains would assist in the proper positioning of the 3′-end of the U6 snRNA within the catalytic site for catalysis.

The KA-1 domain at the C-terminus of TUT1 is conserved among vertebrates (Supplementary Fig. 2). The electrostatic surface potential of the KA-1 domain suggested the presence of a positive charge cluster (Fig. 3e). To evaluate the RNA-binding ability of the KA-1 domain of TUT1, the KA-1 domain itself (amino acids 598–874) was expressed in *E. coli* and purified, and its RNA-binding ability was analysed by gel retardation (Fig. 3f). The $K_d$ value of the KA-1 domain for U6 snRNA-u4 is estimated to be 570 nM. Mutations of the positively charged amino acids to alanines (R871A/K874A or R779A/R783A) reduced the RNA-binding activity. The $K_d$ values of the R871A/K874A and R779A/R783A variants are estimated to be 1,100 and ≫10,000 nM, respectively (Fig. 3f). Moreover, the R779A/R783A mutations in wild-type TUT1 reduced the uridylylation activity (Fig. 3g), and similar to ΔKA-1, the R779A/R783 mutant TUT1 adds two UMPs (Supplementary Fig. 7b). Thus, these analyses confirmed that the previously unidentified C-terminal domain, KA-1, is an RNA-binding domain involved in the U6 snRNA recognition, together with

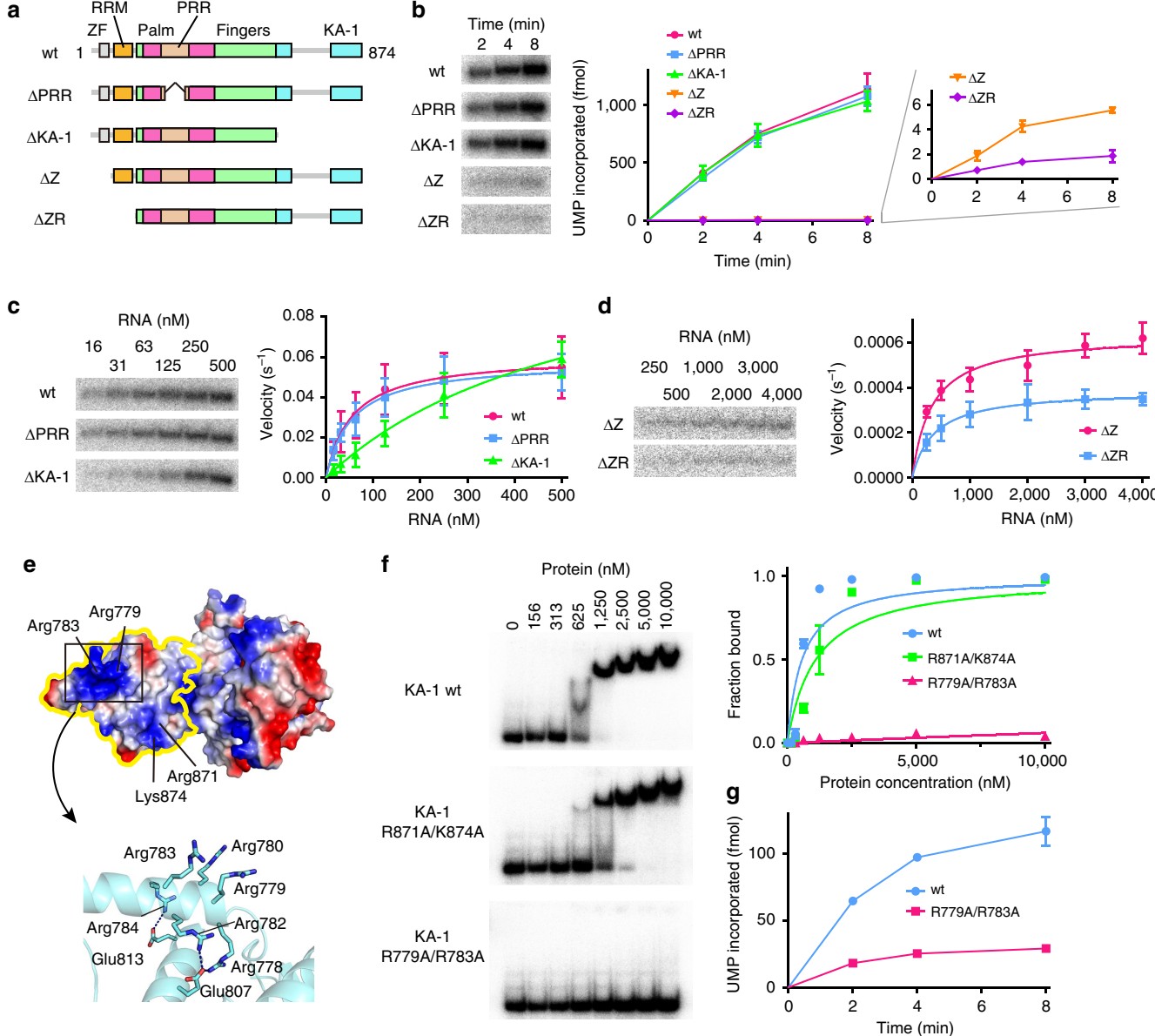

**Figure 3 | Domain requirements for uridylylation by human TUT1.** (**a**) Schematic diagrams of TUT1 variants used for biochemical assays. (**b**) Time-courses of UMP incorporation into U6 snRNA-u4 by wild-type TUT1 and its variants under the standard conditions (1 μM RNA and 10 nM protein). Magnified view of UMP incorporation by ΔZ and ΔZR (right) (**c**) Steady-state kinetics of UMP incorporation into U6 snRNA-u4 by wild-type TUT1, ΔPRR and ΔKA-1, with various RNA concentrations (0–500 nM). (**d**) Steady-state kinetics of UMP incorporation into U6 snRNA-u4 by ΔZ and ΔZR, with various RNA concentrations (0–4,000 nM). (**e**) Electrostatic surface representation of TUT1_ΔN. The positively and negatively charged areas are coloured blue and red, respectively. The KA-1 domain of TUT1 is framed by a yellow line. The positions of R797, R783, R871 and K874 are indicated (upper). A detailed view of the positively charged region in KA-1 (lower). (**f**) Gel retardation assays using various concentrations of wild-type and mutant KA-1 (R871A/K874A and R779A/R783A) proteins (0–10,000 nM). (**g**) Time-courses of UMP incorporation into U6 snRNA-u4 by wild-type and R779A/R783A TUT1 (0.1 μM RNA and 5 nM protein). Bars in the graphs (**b–d**,**f**,**g**) indicate s.d. of two or three independent experiments.

the N-terminal Zn-finger and RRM domains. The different numbers of UMP incorporations between wild-type TUT1 and the R779A/R783A mutant (and ΔKA-1) are discussed below.

As described above, the N-terminal RRM is mobile relative to the catalytic core domains, and the C-terminal domain KA-1 domain rotates relative to the catalytic core domains (Fig. 1c,d). Thus, at the UMP incorporation stage, the domain movements would be coupled with the recognition of the U6 snRNA.

**Interaction between human TUT1 and U6 snRNA.** As TUT1 was originally identified as a U6 snRNA-specific terminal

uridylyltransferase[9], TUT1 tightly interacts with the U6 snRNA *in vivo*[34]. The interactions between the U6 snRNA and TUT1 were analysed by Tb(III) hydrolysis mapping techniques[35], and the protection patterns for the U6 snRNA in the presence and absence of TUT1 were assessed.

In the presence of wild-type TUT1, several significant regions of protection are observed (Fig. 4a–c, Supplementary Figs 8,9). Nucleotides 33–40 and 94–99 are strongly protected by TUT1. These regions correspond to the double-stranded part of the U6 snRNA telestem in the two-dimensional structure (Fig. 4c, left)[7,8]. Nucleotides 59–80, corresponding to the 3′-ISL, and nucleotides 1–20, corresponding to the 5′-stem loop of the U6 snRNA, are

also moderately protected by TUT1. However, in the presence of TUT1, some regions of deprotection are observed around the nucleotides corresponding to the bulge region between the telestem and ISL (nucleotides 48–54, 83–88), the tip of the 3′-ISL (nucleotide 68), and nucleotides 101–104, corresponding to the 3′-end of the U6 snRNA (Figs 4b and 3c left). These observations suggest that the entire region of the U6 snRNA is recognized by TUT1, and that structural changes of the U6 snRNA are induced upon TUT1 binding.

To assess the U6 snRNA recognition by TUT1 in more detail, the interactions between the U6 snRNA and TUT1 variants (ΔZR and ΔKA-1) were further analysed (Fig. 4b,c, Supplementary Figs 8,9). As with wild-type TUT1, both ΔZR and ΔKA-1 protect the telestem region and de-protect the 3′-terminal oligo(U) tail. These observations suggest that the telestem region would be protected by the cleft between the fingers and palm domains, and that the 3′-oligo(U) tail would be unfolded and single-stranded in the catalytic pocket between these domains. ΔZR does not protect the single-stranded region of nucleotides 20–25 (Fig. 4c, middle). In contrast, ΔKA-1 retained the protection of the single-stranded region of nucleotides 20–25, in a similar manner to wild-type TUT1, but did not de-protect the bulged region between the telestem and ISL (nucleotides 48–54, 83–88) and the tip of the 3′-ISL (Fig. 4c, right). These observations suggest that KA-1 would bind the bulged region between the telestem and ISL and induce the conformational change of the tip of the 3′-ISL.

Together, the N-terminal Zn-finger and RRM domains of TUT1 interact with the single-stranded 5′-area of the U6 snRNA, and the KA-1 domain interacts with the bulging loops. These interactions induce the conformational changes of the 3′-ISL and the bulging loop in the U6 snRNA. The core catalytic domain would binds tightly to the double-stranded telestem region, and the 3′-region of the U6 snRNA would be unfolded and single-stranded.

## Discussion

In this study, we determined the crystal structures of human TUT1 and its complexes with nucleotides. The structure of the TUT1 nucleotide-binding pocket is suitable for interactions with UTP, through specific hydrogen bonds between the uracil base and the conserved His and Asn residues in the catalytic pocket (Fig. 2b). TUT1 has a distinct C-terminal domain. The C-terminal domain is topologically homologous to the MARK-3 KA-1 domain, which interacts with phospholipids[26]. The TUT1 C-terminal domain has now been found to function as an RNA-binding domain (Fig. 1a,b). The C-terminal KA-1 domain plays an important role in the uridylylation by TUT1, through its RNA-binding activity (Fig. 3c,f, Supplementary Fig. 7). Almost the entire sequence of the U6 snRNA is recognized by the mobile N-terminal RNA binding domain and the C-terminal KA-1 domain of TUT1, cooperatively with the catalytic core domain, and the recognition of the U6 snRNA by TUT1 is coupled with the domain movements and structural changes in the U6 snRNA (Figs 4 and 5). Thus, the present structural and biochemical studies have demonstrated that TUT1 acts as a U6 snRNA-specific terminal uridylyltransferase.

TUT1 adds several uridines onto the 3′-end of the U6 snRNA in vitro[34] (Supplementary Fig. 7). TUT1 has additional domains at its N- and C- termini, as compared with yeast Cid1, vertebrate mitochondrial PAP (PAPD1) and E. coli PAP[21–24,30,36,37] (Fig. 1a). Since these enzymes lack the additional domain (or region) as compared with TUT1, these enzymes would add multiple nucleotides onto the termini of any RNA. The additional domains in TUT1 would be required for the specific recognition of the U6 snRNA, and allow TUT1 to bind the U6

snRNA on its surface (Fig. 5). Wild-type TUT1, as well as ΔZR and ΔKA-1, protects the telestem and deprotects the 3′-part of the oligo-(U) stretch (Fig. 4c). Thus, the telestem would be relocated to the specific surface between the fingers and palm domains, and the unfolded single-stranded 3′-part of the U6 snRNA would be relocated within the active site, as previously modelled in yeast Cid1 (refs 22,38) (Supplementary Fig. 10).

The presence of N-terminal and C-terminal RNA binding domains would prevent the U6 snRNA from dislodging from the enzyme surface during the uridylylation reaction, by anchoring the entire U6 snRNA molecule (Figs 4,5).

After TUT1 incorporates several UMPs onto the 3′-end of the U6 snRNA by open-to-closed conformation cycles of the catalytic domain[38,39] (Fig. 5), the 3′-part of the oligo-uridylylated tail would be compressed within the active pocket. As a result, the 3′-end of U6 snRNA would no longer be relocated to the active site. Then, TUT1 terminates RNA synthesis and the oligo-uridylylated U6 snRNA is released from the enzyme, as observed in the mechanism for the termination of RNA synthesis by tRNA nucleotidyltransferases[40–43]. This is consistent with the observation that TUT1 mutants, ΔKA-1 or R779A/R783A, add fewer UMPs, as compared with wild-type TUT1 (Supplementary Fig. 7). The absence of KA-1 or the loss of the RNA binding activity of KA-1 would allow U6 snRNA to translocate easily on the enzyme surface, and the U6 snRNA would be released from the enzyme.

Human TUT1 can also function as a polyA polymerase acting on specific mRNAs under oxidative stress conditions[29]. TUT1 interacts with phosphatidylinositol 4-phosphate 5-kinase Iα, PIPKIα and its polyA polymerase activity is also activated by phosphatidylinositol 4,5-bisphosphate (PIns4,5P2) in vitro[29,33]. Upon oxidative stress, TUT1 is recruited into the cleavage and polyadenylation specificity factor (CPSF) complex for the polyadenylation of specific oxidative stress response mRNAs[29,44]. The function of TUT1 as a polyA polymerase is also activated by several protein kinases[31–33].

The complex structure of TUT1 with ATP, presented in this study, showed that the adenine base forms only one hydrogen bond with His549 (Fig. 2c), and the biochemical analysis demonstrated that ATP has lower affinity than UTP in vitro (Fig. 2e). TUT1 can uridylylate the 3′-UTR of the heme-oxygenase-1 (HO1) mRNA transcript (3′-UTR-HO1)[44] in vitro, but the efficiency is much lower than that for the U6 snRNA-u4 transcript, and TUT1 also efficiently incorporates UMP rather than AMP onto the 3′-end of 3′-UTR-HO1 (Supplementary Fig. 11). The interaction of TUT1 with other factors and its phosphorylations by several kinases trigger the CPSF complex formation on specific mRNAs[29,31,33,44,45]. Since the KA-1 domain of MARK-3 binds to phospholipids[26], the mobile KA-1 domain of TUT1 might regulate the PtdIns-4,5-P2-dependent activation of TUT1 as a polyA polymerase. The recruitment of TUT1 into the CPSF complex, its interactions with other factors, and its modifications might induce allosteric structural changes of TUT1, which may lead to changes in the nucleotide-binding pocket structure that make it suitable for the accommodation of ATP, rather than UTP, and allow TUT1 to add polyA tails onto specific mRNAs[44]. The detailed mechanism of the alteration of the nucleotide specificity of TUT1 for specific biological processes awaits further study.

## Methods

**Expression and purification of human TUT1 proteins.** The coding sequence of human TUT1 was synthesized by Eurofins genomics (Japan), with codon-optimization for the expression in E. coli. The nucleotide sequence of the synthetic human TUT1 gene is shown in Supplementary Table 1. For the crystallizations and in vitro biochemical assays, full-length human TUT1 and its variant cDNAs were cloned into

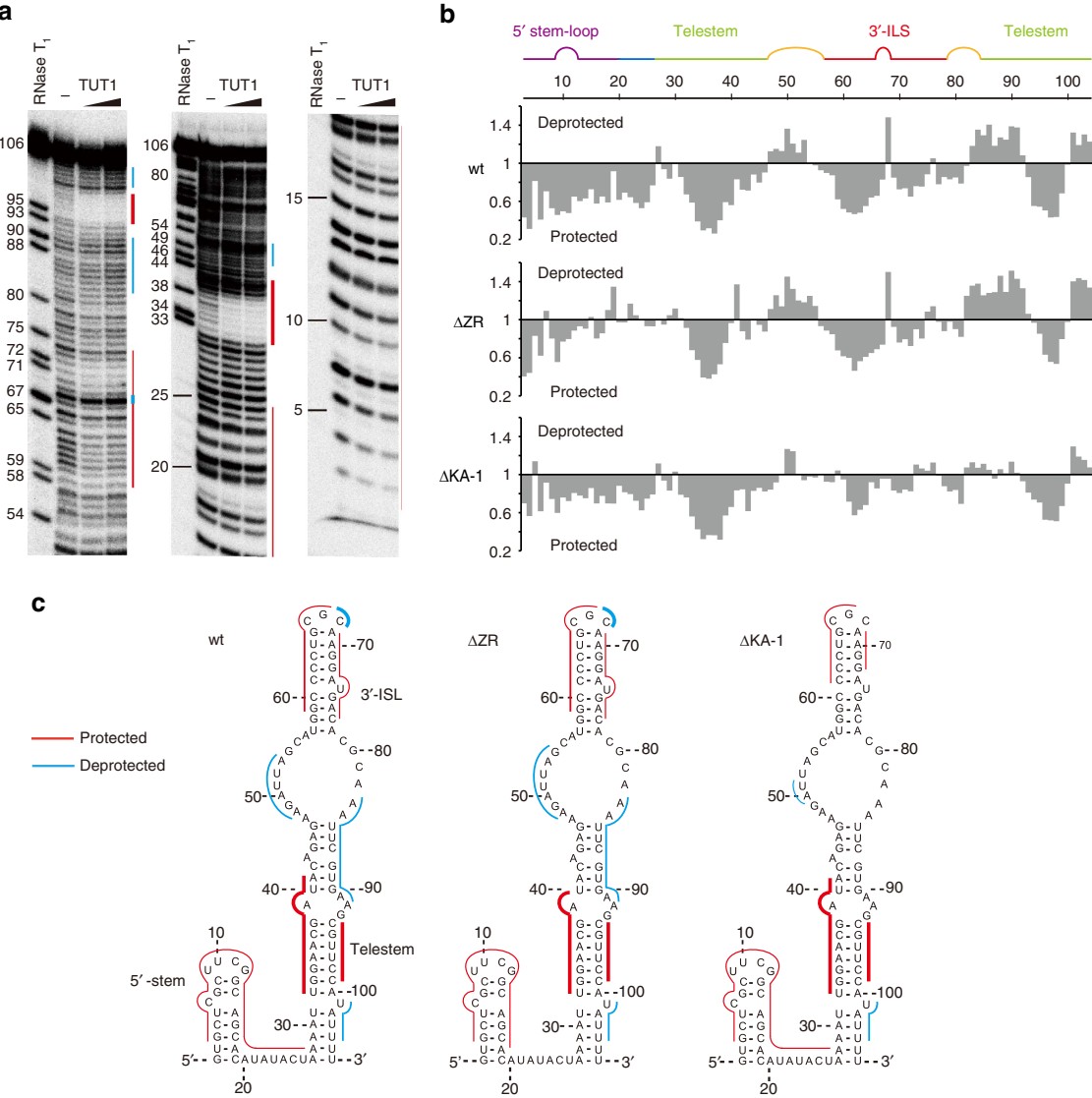

**Figure 4 | Interaction of human TUT1 with U6 snRNA. (a)** Tb(III)-mediated cleavage pattern of U6 snRNA in the absence and presence of TUT1. $^{32}$P-labelled U6 snRNA-u4 was incubated with Tb(III) in absence or presence of 0.4 μM and 0.8 μM recombinant TUT1. Positions of nucleotides were determined by partial digestion of the RNA substrate by RNase T$_1$. Cleavage patterns were analysed on 8% (w/v) (left) or 16% (w/v) (middle and right) sequencing gels. Protected and deprotected regions are depicted by red and cyan lines, respectively. **(b)** Quantitative analysis of TUT1 binding to U6 snRNA. The relative band intensities at each nucleotide position, in the presence and absence of wild-type TUT1 (upper), are shown. Quantitative analysis of ΔZR (middle), and ΔKA-1 (lower) binding, as in wild-type TUT1 (Supplementary Figs 8,9). **(c)** Superimpositions of the footprinting data of wild-type TUT1 (left), ΔZR (middle) and ΔKA-1 (right) onto the secondary structure of the U6 snRNA[34]. Protected and deprotected regions are coloured as in **a**.

the *Nde* I and *Xho* I sites of the pET15b (amino-acid residues 1–702 and 1–748) or pET22b (for the other constructs) vector (Merck Millipore, Japan). The point mutations of cysteine residues were introduced by the overlap PCR method. The nucleotide sequences of primers used for the mutations are shown in Supplementary Table 2. *E. coli* BL21(DE3) (Novagen, Japan) was transformed by the plasmids, and the transformants were grown at 37 °C until the A$_{600}$ reached 1.0. The expression of the TUT1 protein and its variants was induced by adding isopropyl-β-D-thiogalactopyranoside at a 0.1 mM final concentration, and incubating the cultures for 16 h at 18 °C. The cells were collected, and lysed by sonication in buffer containing 20 mM Tris-HCl, pH 7.0, 500 mM NaCl, 10 mM β-mercaptoethanol, 20 mM imidazole, 0.1 mM phenylmethylsulfonyl fluoride and 5% (v/v) glycerol. The proteins were first purified by chromatography on a Ni-NTA agarose column (QIAGEN, Japan), and then further purified on a HiTrap Heparin column (GE Healthcare, Japan). Finally, the proteins were purified by chromatography on a HiLoad 16/60 Superdex 200 column (GE Healthcare, Japan), in buffer containing 20 mM Tris-HCl, pH 7.0, 400 mM NaCl and 10 mM β-mercaptoethanol. The purified proteins were concentrated and stored at − 80 °C until use.

**Crystallization and structural determination of TUT1.** All of the crystals used for the structural determination were prepared by the hanging drop vapour

diffusion method at 4 °C. The protein concentrations were adjusted to 5 mg ml$^{-1}$ before use, and 1 μl of protein solution was mixed with 0.5 μl of reservoir solution. For the crystallization of the form-I and -II crystals, TUT1_ΔN with the C372A/C415A/C501A/C504S and C372A/C399A/C415A/C501A/C504S/C574A mutations were used. The protein solution (5 mg ml$^{-1}$ concentration) was supplemented with 10 mM DTT and 100 mM sodium acetate before use. The protein solution was mixed with the reservoir solution containing 100 mM Hepes-NaOH, pH 7.5, 13–15% (w/v) PEG3350 and 2–4% (w/v) tacsimate, pH 8.0. For the crystallization of the form-III crystals, TUT1_ΔN with the C501A/C504S mutation was used. The protein solution was mixed with the reservoir solution, containing 100 mM sodium cacodylate, pH 7.0, 12% (w/v) PEG4000, and 8% (w/v) tacsimate, pH 8.0. For the crystallization of the form-IV crystals, TUT1_ΔC with the C372A/C399A/C415A/C501A/C504S/C574A mutations was used. The protein solution was mixed with the reservoir solution, containing 15% (w/v) PEG3350, 100 mM Tris-HCl, pH 7.0 and 200 mM MgCl$_2$.

The data sets were collected at beamlines 17A and NW12A at the Photon Factory at KEK, Japan. The crystals were flash-cooled with the reservoir solution supplemented with cryoprotectant: 30% (v/v) ethylene glycol for the form-I and -II crystals, 30% (v/v) sucrose and 5% (v/v) ethylene glycol for the form-III crystals

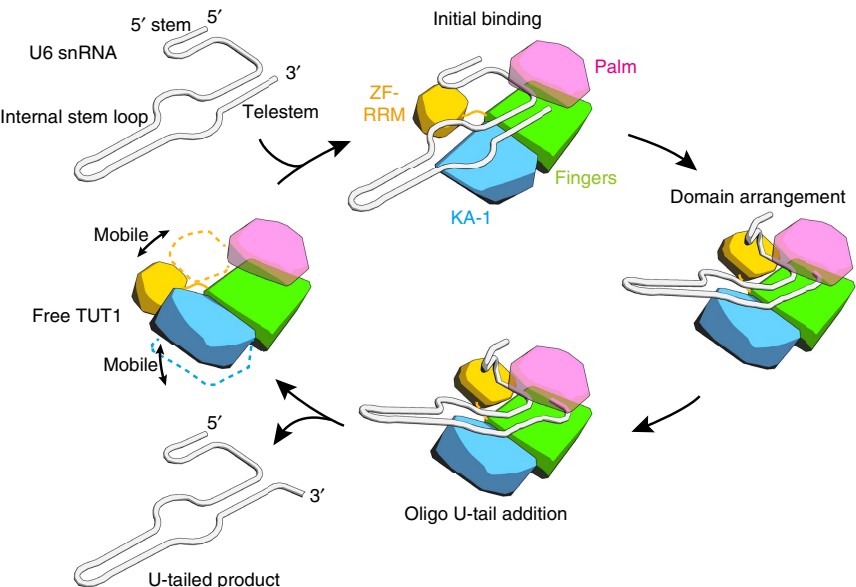

**Figure 5 | Oligo-uridylylation of the 3′-end of U6 snRNA by TUT1.** Schematic diagram of the oligo-uridylylation of the 3′-end of U6 snRNA by human TUT1. The N-terminal Zn-finger and RMM, palm, fingers and KA-1 domains are coloured orange, magenta, green and cyan, respectively. The Zn-finger, RMM and KA-1 are mobile, relative to the catalytic palm and fingers domains, and interact with specific regions within the U6 snRNA.

and 30% (v/v) sucrose for the form-IV crystals. For the data collection of the MgUTP- and BaUTP-bound complexes, the cryoprotectants were supplemented with 2 mM UTP and 4 mM $MgCl_2$ and 2 mM UTP and 4 mM $BaCl_2$, respectively, and the crystals were incubated for 30 min at room temperature before data collection. For the data collection of the MgATP-bound complex, the cryoprotectant was supplemented with 10 mM ATP and 20 mM $MgCl_2$, and the crystals were incubated for at least 6 h at 4 °C. The data were indexed, integrated and scaled with XDS[46]. The diffraction data of the form-III *apo* crystal, which exhibited strong anisotropy, were anisotropically scaled[47]. The initial phase was determined by the SIRAS method, with the signal from the barium ions coordinated by the UTP and aspartate residues (Supplementary Fig. 3). The heavy atom sites were determined by SHELX[48] with the hkl2map interface[49]. The phase was calculated and density-modified by SOLVE/RESOLVE[50,51]. The data sets of the *apo* and BaUTP-bound crystals in form-I were used as the native and the derivative, respectively. The structures were refined with phenix.refine[52], and manually modified with Coot[53]. The representative images of the electron density are shown in Supplementary Fig. 12

**In vitro nucleotide transferase assay.** RNA substrates (U6 snRNA-u4 and 3′-UTR-HO1) were synthesized by T7 RNA polymerase, using plasmids encoding the respective template DNA sequences downstream of the T7 promoter, and were purified by 10% (w/v) polyacrylamide gel electrophoresis under denaturing conditions. The nucleotide sequences are 5′-GUGCUCGCUUCGGCAGCACAUA UACUAAAAUUGGAACGAUACAGAGAAGAUUAGCAUGGCCCCUGCGCA AGGAUGACACGCAAAUUCGUGAAGCGUUCCAUAUUUU-3′ for U6 snRNA-u4, and 5′-GGGUUUUUAUAGCAGGGUUGGGGU GGUUUUUUGAGC CAUGCGUGGGUUGGGGAGGGAGGUGUUUAACGGCACUGUGGCCUUGGU CUAACUUUUGUGUGAAAUAAUAAACAACAUUGUCU-3′ for 3′-UTR-HO1.

For UMP (or AMP) incorporation into U6 snRNA-u4 or 3′-UTR-HO1, 20 μl reaction mixtures, containing 20 mM Tris-HCl, pH 8.5, 100 mM NaCl, 10 mM $MgCl_2$, 1 mM DTT, 100 μM UTP (or ATP), 33 nM α-$^{32}$P UTP (or ATP) (3,000 Ci per mmol; Perkin Elmer, Japan), 1 μM RNA transcript and 10 nM TUT1 (or its variants), were incubated at 37 °C (standard assay conditions). At the indicated time points (2, 4 and 8 min), a 5 μl portion of the reaction mixture was withdrawn and the reactions were stopped. The $^{32}$P-labelled RNAs were separated by 10% (w/v) polyacrylamide gel electrophoresis under denaturing conditions. The intensities of the $^{32}$P-labelled RNAs were quantified with a BAS-5000 imager (Fuji Film, Japan). To compare the activities of wild-type TUT1 and the R779A/R783A mutant, 67 nM α-$^{32}$P UTP, 100 nM RNA and 5 nM enzyme were used in the reaction mixture.

For the determination of the $K_m$ and $k_{cat}$ values of UTP, reaction mixtures containing 20 mM Tris-Cl, pH 8.5, 100 mM NaCl, 10 mM $MgCl_2$, 1 mM DTT, various concentrations of UTP (16–500 μM, 300 mCi per mmol), 10 nM TUT1 and 1 μM U6 RNA-u4 were incubated at 37 °C for 2 min. The $K_m$ and $k_{cat}$ values of ATP were determined with modifications, using 63–2,000 μM of ATP (150 mCi per mmol) and 100 nM TUT1 in the reaction, in the same buffer conditions as those used for the determination of the $K_m$ and $k_{cat}$ values of UTP. For the determination

of the $K_m$ and $k_{cat}$ values of U6 snRNA-u4 by wild-type TUT1, ΔPRR and ΔKA-1, reaction mixtures containing 20 mM Tris-HCl, pH 8.5, 100 mM NaCl, 10 mM $MgCl_2$, 1 mM DTT, 500 μM UTP, 33 nM α-$^{32}$P UTP, various amounts of U6 snRNA-u4 (16–500 μM) and 5 nM enzymes were incubated at 37 °C for 2 min. The $K_m$ and $k_{cat}$ values of U6 RNA-u4 by ΔZ and ΔZR were determined with several modifications, using 250–4,000 nM of U6 RNA-u4 and 100 nM ΔZ and ΔZR in the reaction, in the same buffer conditions as those used for the determination of the $K_m$ and $k_{cat}$ values of wild-type TUT1.

For the analysis of the number of UMP incorporations onto the 3′-termini of U6 snRNA-u4 by wild-type TUT1 and its variants, 50 μl reaction mixtures, containing 20 mM Tris-HCl, pH 8.5, 100 mM NaCl, 10 mM $MgCl_2$, 1 mM DTT, 500 μM UTP, 50 nM RNA transcript and 50 nM TUT1 (or its variants), were incubated at 37 °C. At the indicated time points (2, 5, 10 and 15 min), a 10 μl portion of the reaction mixture was withdrawn and the reactions were stopped. The reaction products were separated by 10% (w/v) polyacrylamide gel electrophoresis under denaturing conditions. The gels were stained with ethidium bromide.

Uncropped images of the scans and gels are shown in Supplementary Figs 13 and 14.

**RNA footprinting with Tb(III).** RNA footprinting with terbium chloride (TbIII) was performed[35,54] as follows. Increasing amounts of recombinant TUT1 or its variant proteins (0.4 μM and 0.8 μM for full-length (1–874), 0.8 μM and 1.6 μM for ΔKA-1 (1–702), 1–748, and 54–874, and 1.6 μM and 3.2 μM for ΔN (141–874)) were preincubated with 60 nM of 5′-[$^{32}$P]-labelled U6 snRNA-4u transcript, in buffer containing 50 mM Tris-HCl, pH 8.5, 100 mM NaCl, 10 mM $MgCl_2$ and 1 mM DTT, for 10 min at 22 °C. Tb(III) was then added to a final concentration of 12.5 mM. The mixture was incubated for 2 h, and the cleavage reaction was quenched by the addition of 50 mM EDTA, pH 8.0 and 0.5% (w/v) SDS with loading buffer containing 9 M urea. The products were separated by 8 or 16% (w/v) polyacrylamide gel electrophoresis under denaturing conditions. The cleavage patterns were analysed with a BAS-5000 imager, and the band intensities were quantified with the Image Gauge software, Ver. 4.0 (Fuji Film, Japan). The raw quantification data of the band intensities are provided (Supplementary Fig. 9). Each peak was assigned to the corresponding nucleotide in U6 snRNA, and the band intensities were defined as the integrated areas of the peaks. The band intensities in the presence of TUT1 or its variants were divided by the band intensities in the absence of proteins. The relative intensities of cleavage were mapped onto U6 snRNA. For the protection/deprotection criteria in the presence of TUT1 and its variants, the relative band intensities > 1.0 and < 1.0 were interpreted as deprotected and protected regions in the RNA, respectively. The relative deprotected and protected band intensities were expressed with plus and minus values, respectively.

**Electrophoretic mobility shift assay.** For the electrophoretic mobility shift assay, increasing amounts of recombinant KA-1 were incubated with 1 nM of 5′-$^{32}$P-labelled U6 RNA-u4, in buffer containing 50 mM Tris-HCl, pH 8.5, 100 mM

NaCl, 10 mM MgCl₂, 10 mM β-mercaptoethanol and 10% (v/v) glycerol, for 15 min at room temperature. The samples were separated by 6% (w/v) polyacrylamide electrophoresis under native conditions. The intensities of the $^{32}$P-labelled RNAs were quantified with a BAS-5000 or FLA-3000 imager (Fuji Film, Japan).

**Data availability**. Coordinates and structure factors have been deposited in the Protein Data Bank, under the accession codes 5WU1, 5WU2, 5WU3, 5WU4, 5WU5 and 5WU6. All other data are available from the corresponding author upon reasonable request.

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

## Acknowledgements

We thank Azusa Hamada for technical assistance. We thank the beamline staff of BL-17A and NW12A (KEK, Tsukuba) for technical assistance during data collection. This work was supported in part by grants from the Funding Program for Next Generation World-Leading Researchers of JSPS (to K.T.), Grants-in-Aid for Scientific

Research (A) (to K.T.), Grants-in-Aid for Young Scientists (B) (to S.Y.), and a Grant-in-Aid for Scientific Research on Innovative Areas (to K.T.) from JSPS, as well as grants from Takeda Science Foundation (to K.T.), Astellas Foundation for Research on Metabolic Disorders (to K.T.) and The Hamaguchi Foundation (to K.T.).

## Author contributions

K.T. planned and designed the research; S.Y. performed the purification, crystallization and structure determination of TUT1. S.Y., T.N., Y.T. and K.T. performed biochemical assays. S.Y. and K.T. analysed the data, and wrote the paper.

## Additional information

**Competing interests:** The authors declare no competing financial interests.

