## [Peer review file · Nature Communications]

Reviewers' comments:

Reviewer #1 (Remarks to the Author):

Yamashita et al. describes the crystal structure of human terminal uridylyl transferase which catalyses uridylylation of the 3'-end of U6 spliceosomal snRNA. Unlike other spliceosomal snRNAs U6 snRNA is transcribed by RNA polymerase III and has a 2'-3' cyclic phosphate at the terminus. U6 snRNA isolated from HeLa cells have variable number of uridylylates which are added by TUT1. Gu et al. (J. Biol. Chem. 272, 21989-21993 (1997)) showed that the terminal phosphate is derived from the alpha-phosphate of post-transcriptionally added uridylylate so this process involves removal of uridine which leaves 2'-3' cyclic phosphate, decyclization, dephosphorylation and UMP addition by TUT1 and removal of uridine to create a new 2'-3' phosphate 3'terminus. The uridylylate tail of U6 snRNA is the binding site for Lsm protein complex. Biologically important questions to address are (1) what determines its specificity for U6 snRNA; (2) how processive TUT1 is and how the number of uridylylate added is regulated; (3) how the specificity for UTP is determined.

In addition to the catalytic core domain, which consists of the Palm (with the proline-rich repeat embedded) and Finger domains, it has a putative Zn-finger and RRM at the N-terminus and the kinase associated-1 (KA1) domain at the C-terminus. Although the full-length protein did not yield any crystals, the authors managed to crystal truncated protein. Crystal form I-III lacks the N-terminal Zn-finger and RRM as well as the proline-rich repeat (PRR). Crystal form IV lacks the N-terminal Zn-finger and the C-terminal KA-1 domain as well as PRR. TUT-1 also has a poly-A polymerase activity and the authors determined the structure of TUT-1 with UTP and ATP and showed how these nucleotides are recognised by TUT-1. For these assay U6 snRNAu4 substrate was used and the authors investigated how the substrate is recognised by various domain of TUT-1 using RNA chemical protection, TUT1 enzymatic activity assay and electrophoretic mobility shift assays.

The authors should run the products of uridylylation reaction by TUT1 and its derivatives on high resolution gel to show how many UMPs are added to the termini.

Crystallographic work was carried out to high technical quality. It would be helpful if Figure panels 2b and 2c are larger so that readers can see how UTP and ATP are recognised by surrounding residues in TUT1. It is important to visualise Mg⁺⁺ and water molecules. Have the authors include oligo-U as an acceptor?

Fig 3b shows that the deletion of Zinc-finger or Zinc-finger plus RRM abolish UMP incorporation to U6 snRNA-u4 but the deletion of the Pro-rich repeat or KA-1 domain had no effect. Data for ΔF and ΔZR are overlapping in the right hand panel of Figure 3b. It should be made clear. Fig 3c shows that KA1 domain does not affect U6 snRNA u4 binding substantially but Fig 3f shows that KA-1 domain does bind U6 snRNA-u4 and the mutations of R779 and R783 abolish RNA binding. The authors should clarify this point. It would be interesting for the authors to carry out UMP incorporation assay with TUT1 with these mutations.

Fig 3a shows that the deletion of the Zinc-finger alone abolishes substrate RNA binding but is the RRM contributing to substrate binding? It would be help if the authors show

electrostatic surface potential of RRM to see it is electropositive.

In RNA protection assay the RNA should not be cut more than once as the first cleavage may induce structural change in RNA and the second cleavage does not reflect the structure of the uncleaved RNA. Hence more than 90 % of RNA should remain uncleaved. It is not clear to this referee how the authors estimated the band intensity. Is it normalized to free RNA? How did the authors translate the data in panel b to panel c. In panel b 5'stem-loop is not well protected by ΔZR , moderately protected by $\Delta KA-1$ and well protected by the wild type. Did the authors applied some quantitative criteria to decide whether RNA is protected, deprotected or unchanged? Is the observed difference sufficient to say which domain interacts with which region of RNA? It is not clear how the authors come up with a model proposed in Fig 5.

I think the manuscript describes experiments interesting to reader of Nature communications but before the manuscript is considered further the authors should address the points described above.

Reviewer #2 (Remarks to the Author):

TUT1 (also known as U6 TUTase, RBM21 and Star-PAP) was previously purified on the basis of its ability to perform the 3' end modification of spliceosomal U6 snRNA. Newly synthesized U6 snRNA contains four UMP residues at its 3' end, but undergoes shortening at this end if it is not stabilized by terminal modification to form a 2', 3'-cyclic phosphate. TUT1 restores the authentic length of U6 snRNA molecules by terminal addition of up to four UMP residues. Curiously, TUT1 has also been suggested by one other group to be a poly(A) polymerase (Star-PAP) involved in the regulated 3' end processing of specific target mRNAs in a manner regulated by nuclear phosphoinositides. Thus the biochemical activity and biological roles of this enzyme have remained rather contentious for a number of years, awaiting detailed characterization.

In this rigorous study, Yamashita and colleagues present compelling structural and biochemical evidence in support of the view that TUT1 is a U-specific terminal nucleotidyl transferase that selectively binds its U6 substrate via interactions with multiple RNA-interaction domains. In particular, the strong preference of the enzyme for UTP over ATP in vitro raises important questions about the way in which TUT1 could possibly act as a poly(A) polymerase in vivo, as reported by another group. In addition, the C-terminal KA-1 domain, reported by the other group to bind phospholipids, is shown here to have RNA-binding activity. The data are of a high overall quality and the manuscript is clearly and concisely written; I can suggest no modifications. Given the previous controversy about the activity of TUT1, and the extent to which this is resolved by this definitive study, it is in my view important that the manuscript is published in a prominent journal.

Reviewer #3 (Remarks to the Author):

Yamashita et al., present the X-ray crystal structures of two shortened variants of human TUT1 along with biochemical data for the wild-type enzyme and domain mutants.

The work is timely and interesting. However, it should be improved on a number of points before it is suitable for publication.

1. Given the very low activity of delta-Z and delta-ZR in the current work, the two crystallized constructs, TUT1_deltaN and TUT1_deltaC, are most probably catalytically dead. However, this should be tested. I don't find activity tests of these constructs described or shown in the present manuscript.
2. The interaction of the wt-KA-1 domain with RNA is shown in Fig. 3f and the protection analysis of the wt, deltaZR and the deltaKA-1 construct is informative. However, once again the crystallised constructs have not been analysed. The results of these experiments govern how much trust one can put into the scheme in figure 5 and parts of the discussion.

Minor:

p. 10, line 10 "reportedly" as used here signals to most readers that the authors doubt the results in ref. 26. Either describe what is doubtful (with references etc.) or remove reportedly.

p. 10 line 19-20. The wording here gives the impression that U6 snRNA has been tested as a substrate for the other enzymes. The authors want probably to contrast the putative TUT1 mechanism to these other enzymes. Rewrite, for clarity, these lines together with the following two lines (21-22).

p. 10 lines 25-27. I don't find any evidence in the current manuscript for the RNA binding specificity described here for the active site and the area between the fingers and palm domain. There are no references here for other work either. If pure speculation, fine, but then it should be noted as such.

p. 11 lines 1-5. What is the evidence for that TUT1 terminates catalysis due to a "full" nucleotide binding site?

Reviewer #1

Thank you for your valuable and constructive comments and suggestions for improving our manuscript. According to the comments and suggestions, we provided additional experimental data and supplementary figures, and revised the manuscript.

1) According to the suggestion, we ran the reaction products of uridylylation by wild-type TUT1 and its variants on a higher resolution gel (Supplementary Fig. 7 in the revised manuscript). The results confirmed that TUT1 has oligo-uridylylation activity. The results also showed that wild-type and Δ PRR rapidly add three to four UMPs onto the termini of U6 snRNA-u4. In contrast, Δ KA-1 adds two to three UMPS under the same conditions. The oligo-uridylylation activity of TUT1 and the different activities between wild-type TUT1 and Δ -KA1 were described clearly and discussed in the text (Page 8, line 22-, Page 9, line 10-, Page 12 line 3- in the revised manuscript), by referring to Supplementary Fig. 7.

2) According to the suggestion, we provided stereo-view panels of Figure 2b and 2c in the revised manuscript, so that readers can readily understand the UTP or ATP recognition and the Mg^{2+} coordination in the TUT1 catalytic pocket. We have not determined the structure of TUT1 complexed with oligo-U as an acceptor primer in this study.

In the revised manuscript, the superimposition of the structure of TUT1 complexed with UTP with the yeast Cid1 structure complexed with ApU was presented (Supplementary Fig. 6 in the revised manuscript). The superimposition confirmed that, in the complex structure of TUT1 with UTP (or ATP), the nucleotide binds to the incoming nucleotide binding site. The positions of the nucleotide binding in the incoming nucleotide binding site, shown in Fig. 2a and 2c, was described by referring to the Supplementary Fig. 6 (Page 7 line 10-).

3) According to the suggestion, the part of the graph (Figure 3b), showing the time course of UMP incorporations into U6 snRNAu4 by Δ Z and Δ ZR was enlarged (Figure 3b inset).

According to the steady-state kinetics of UMP incorporations onto 3-termini of U6 snRNA-u4 by Δ KA-1 (Figure 3c), the K_m values of Δ -KA1 toward U6 snRNA-u4 were about ten-fold higher than that of wild-type TUT1, suggesting that the presence of KA-1 increases the affinity of TUT1 toward U6 snRNA-u4. The gel-retardation assays using the isolated KA-1 domain and its mutant variants showed that the isolated KA-1 domain itself has RNA-binding ability. The mutant KA-1 (R779A/R783A) lost its RNA-binding ability. Since U6 snRNA is recognized by the domains (ZN, RRM, palm, fingers and KA-1) of TUT1 cooperatively, the deletion of KA-1 from TUT1 does not completely abolish the activity. These points are clearly described in the revised manuscript (Page 9 line 10 -).

Furthermore, according to the suggestion, we tested the oligo-uridylylation activity using the TUT1 mutant (R779A/R783A), and showed that the mutant TUT1 reduced the

uridylylation activity, as compared with wild-type TUT1 (Fig. 3g). This is described in the revised manuscript (Fig. 3g, Page 9, line 10 -). The results further support our proposal that the KA-1 domain of TUT1 is indeed the RNA binding domain.

4) According to the suggestion, we described more details of the RRM of TUT1, and showed the electrostatic surface potential of the RRM of TUT1 (Page 6, line 13 -, Supplementary Fig. 5). A single stranded region of the U6 snRNA would be recognized by the surface of the anti-parallel β -strand of RRM of TUT1, using the conserved aromatic and hydrophobic residues, as observed in other RRMs.

5) According to the comment, we clearly described the methods for the quantification of the RNA protection assays in Figure 4 in the revised manuscript (Methods: Page 24, line 24 -).

The band intensities in the gels were quantified by using the Image Gauge software, Ver. 4.0 (Fuji Film, Japan). The raw quantification data of the band intensities are provided in Supplementary Figs. 8, 9 in the revised manuscript. The band intensities in the presence of TUT1 and its variants were normalized by the respective band intensities in the absence of proteins. The relative intensities of cleavage were mapped onto the U6 snRNA, as shown in Figure 4b. For the protection/deprotection criteria in the presence of TUT1 and its variants, the relative band intensities > 1.0 and < 1.0 were interpreted as deprotected and protected regions in RNA, respectively. The relative deprotected and protected band intensities were expressed with plus and minus values, respectively, as shown in Figure 4b.

Since majority of RNA remain uncleaved (Supplementary Fig. 8, 9), it is likely that the majority of cleavage patterns reflect the first cleavage of RNA. Even the second cleavage occurred; the cleavage patterns in the presence of proteins are interpreted by subtracting the cleavage patterns in the absence of proteins. Therefore, the comparison of the protection/deprotection patterns in the presence of wild-type TUT1 and its variants is sufficient to determine which domain of TUT1 interacts with which region of U6 snRNA.

Reviewer #2

Thank you for your favorable comments on our manuscript.

Reviewer #3

Thank you for valuable and constructive comments and suggestions for improving our manuscript. According to the comments and suggestions, we provided additional experimental data and Supplementary Figures.

1) According to the suggestions, we tested the uridylylation of U6 snRNA-u4 by TUT1_ΔN and TUT1_ΔC, used for crystallization, in the revised manuscript (Supplementary Fig. 4). As expected, these constructs were almost completely catalytically inactive. We described the activities of these proteins (Page 5, line 3 -).

2) We performed the biochemical reactions and protection assay using the same constructs as described in the first manuscript (Figs. 3 and 4). Unfortunately, the TUT1 variants used for the biochemical reactions and protection assays did not generate diffractive crystals. We do not believe that the protection assay with the TUT1 variants used for the crystallization would provide additional deeper advancements toward understanding U6 snRNA recognition by TUT1.

3) According to the suggestion, we removed the word “reportedly” and rewrote the sentences in the revised manuscript (Page 11, line 3 -).

4) According to the comment, we described the RNA binding specificity in the active site and the area between the finger and palm domains in the revised manuscript (Page 11, line 21 - in the revised manuscript).

Since the bottom of the telestem region is deprotected in the presence of TUT1 (and even in the presence of Δ ZN and Δ KA-1), it is likely that the bottom of the telestem of U6 snRNA would be unfolded in the active site, between the palm and finger domains. Based on the previously reported structure of yeast Cid1 and the RNA path model, along with the RNA protection assays (Figure 4), we have built the RNA path model in the active site and the area between the palm and finger domains of TUT1. In this model, the single-stranded 3'-part of U6 snRNA enters the active site (Supplementary Fig. 10). We clearly described that the bottom of the telestem would be unfolded in the active site, the single stranded termini would enter the active site between the palm and finger domains, and the telestem would interact with the cleft between the palm and fingers (Page 11, line 21). We also cited the references of the previous RNA-path model of yeast Cid1.

5) According to the comment, we clearly rewrote the comparison of the mechanisms of TUT1 and other enzymes in the revised manuscript (page 11, line 15 -).

Since yeast Cid1, vertebrate mitochondrial PAP (PAPD1) and *E. coli* PAP lack the additional domain (or region), as compared with TUT1, these enzymes would add multiple nucleotides onto the termini of any RNA.

6) According the comment, we rewrote the possible mechanism of the termination of uridylylation of RNA by TUT1 in the revised manuscript (page 11, line 30 -).

It is likely that after the oligo-uridylylation of the termini of RNA, the termini of the uridylylated RNA would not be located in the active site, with the U6 snRNA bound on the surface of TUT1 through multiple domains-RNA interactions. This mechanism is similar to those observed in tRNA nucleotidyltransferases, such as CCA-adding enzymes.

REVIEWERS' COMMENTS:

Reviewer #1 (Remarks to the Author):

The authors addressed all the points satisfactorily and the manuscript is considerably improved. I am happy to recommend the revised manuscript for publication.

Reviewer #3 (Remarks to the Author):

I find the manuscript acceptable for publication in its revised form. I congratulate the authors to work well performed.